# Pluripotent Stem Cell-Based Models: A Peephole into Virus Infections during Early Pregnancy

**DOI:** 10.3390/cells9030542

**Published:** 2020-02-26

**Authors:** Claudia Claus, Matthias Jung, Judith M. Hübschen

**Affiliations:** 1Institute of Virology, University of Leipzig, Johannisallee 30, 04103 Leipzig, Germany; 2University Clinic and Outpatient Clinic for Psychiatry, Psychotherapy, Psychosomatic Medicine, Martin Luther University Halle-Wittenberg, 06112 Halle (Saale), Germany; matthias.jung@uk-halle.de; 3Infectious Diseases Research Unit, Department of Infection and Immunity, Luxembourg Institute of Health, 4354 Esch-sur-Alzette, Luxembourg; Judith.Huebschen@lih.lu

**Keywords:** teratogenesis, embryonal development, interferon, placenta, blastocyst, iPSC, pluripotent stem cells, organoid, cytomegalovirus, Zika virus, rubella virus, congenital virus infection

## Abstract

The rubella virus (RV) was the first virus shown to be teratogenic in humans. The wealth of data on the clinical symptoms associated with congenital rubella syndrome is in stark contrast to an incomplete understanding of the forces leading to the teratogenic alterations in humans. This applies not only to RV, but also to congenital viral infections in general and includes (1) the mode of vertical transmission, even at early gestation, (2) the possible involvement of inflammation as a consequence of an activated innate immune response, and (3) the underlying molecular and cellular alterations. With the progress made in the development of pluripotent stem cell-based models including organoids and embryoids, it is now possible to assess congenital virus infections on a mechanistic level. Moreover, antiviral treatment options can be validated, and newly emerging viruses with a potential impact on human embryonal development, such as that recently reflected by the Zika virus (ZIKV), can be characterized. Here, we discuss human cytomegalovirus (HCMV) and ZIKV in comparison to RV as viruses with well-known congenital pathologies and highlight their analysis on current models for the early phase of human development. This includes the implications of their genetic variability and, as such, virus strain-specific properties for their use as archetype models for congenital virus infections. In this review, we will discuss the use of induced pluripotent stem cells (iPSC) and derived organoid systems for the study of congenital virus infections with a focus on their prominent aetiologies, HCMV, ZIKV, and RV. Their assessment on these models will provide valuable information on how human development is impaired by virus infections; it will also add new insights into the normal progression of human development through the analysis of developmental pathways in the context of virus-induced alterations. These are exciting perspectives for both developmental biology and congenital virology.

## 1. New Perspectives for Congenital Virology

The rubella virus (RV) was not only the first human teratogen identified, and as such the first human pathogenic virus that has been classified as a teratogen, it is still one of the most efficient teratogenic viruses. A teratogen is defined as a physical, chemical, or infectious agent associated with physical or functional birth defects, including growth retardation and pregnancy loss, that result from abnormal embryonal or fetal development (www.embryo.asu.edu/handle/10776/7510). Historically, malformations associated with congenital RV infection, as first described by Sir Norman Gregg in 1941, have formed our current concept of teratogenicity [1]. In 1959, Wilson postulated the six principles of teratology, which are still valid today (embryo.asu.edu/handle/10776/7893). The second principle describes that susceptibility to teratogenesis depends on the developmental stage at the time point of exposure to the teratogen. Thus, our understanding of teratogenic mechanisms caused by agents such as RV is strictly dependent on our knowledge of human development, which has undergone some paradigm shifts in recent years. The first paradigm that has been challenged is the all-or-none hypothesis. It states that exposure to embryotoxic or teratogenic agents before organogenesis either does not affect embryonal development at all or results in embryonic death. However, cell death as a requirement for this hypothesis is not a general consequence of exposure to these agents. Instead, their mode of action and the associated congenital malformations are rather developmental stage-specific [2]. Another paradigm shift refers to our view of pregnancies as a state of immune suppression. This is now replaced by our new understanding of the placenta and decidua as immunologically active organs [3]. The elicited antiviral immune response mechanisms include interferon (IFN) signaling as a very efficient first line of defense against pathogenic viruses. The decidua as the maternal compartment of this embryo/fetal-maternal-interface results from morphogenetic restructuring of the endometrium as the inner lining of the uterine wall. Moreover, pregnant women have the capacity to elicit a solid immune response [3], and the fetus itself is not entirely dependent on maternal immune functions. On the contrary, the maternal antiviral countermeasures are supported by the fetal immune response. This was revealed by a mouse model for the congenital Zika virus (ZIKV) infection based on the heterozygous knock-out of the type I IFN receptor, IFNAR, as a result of the crossing of IFNAR^-/-^ female mice with wild-type males [4]. Thus, in pregnant dams, IFNAR^-/+^ fetal cells in the placenta were facing IFNAR^-/-^ cells in maternal tissue. In comparison to their homozygous (IFNAR^-/-^) counterparts, the placental damage caused by ZIKV infection was reduced. Moreover, these heterozygote (IFNAR^-/+^) pups were partially protected from high viral burden, especially in the brain [4]. This leads to the third paradigm shift, the contribution of the IFN system to the protection of embryonal development. Compared to somatic cells, IFN signaling components are attenuated in embryonal and induced pluripotent stem cells (ESCs and iPSCs, respectively) as a cell culture model for early human development [5]. The generation of iPSCs, through transfer of a cocktail of four pluripotency factors into human dermal fibroblasts using retroviral vectors as an ESC-counterpart with less ethical constraints, has changed developmental biology tremendously [6]. The attenuation of the innate immune response in pluripotent stem cells appears to be due to the severe consequences IFN-associated inflammation could have for embryonal development. These negative effects would outweigh its beneficial antiviral activity. An engineered type I IFN response impaired, not only the morphology and pluripotency of iPSCs, but also their endodermal differentiation capacity [7]. Thus, other mechanisms of protection against vertically transmitted pathogens appear to be in place during the very early phase of embryogenesis. One protective mechanism is provided by the intrinsic expression of a subset of IFN-stimulated genes (ISGs) including members of the interferon-induced transmembrane (IFITM) family at high levels, which in somatic cells are only expressed at the basal level and require IFN for their activation [5]. This highlights that our concept of human embryogenesis, and as such of congenital virus infections as initiated in 1941 by Sir Norman Gregg on malformations associated with maternal RV infections [1], is in a constant state of flux. Both research areas, developmental biology and congenital virology, could mutually enrich each other. Not only does the analysis toolbox for virus-associated teratogenic alterations during embryonal development continuously advance, but teratogenic virus infections could also be used for the validation of these newly established models for embryonal development. The well-characterized symptoms associated with congenital virus infections can be correlated to developmental pathways and signaling circuits, and vice versa. Transplacental transmission, the crosstalk at the embryo/fetal-maternal compartment, and the mode of impairment of early embryonal development, remain ill-defined, not only for RV. Although there are already a number of recent reviews on RV and its associated teratogenicity, which is strictly human specific, the potential value of RV for the analysis of congenital virus infections in pluripotent stem cell-based models for the early steps of human development has not yet been discussed. These models offer insights into so far inaccessible processes during very early embryonal development and, thus, complement animal models and overcome some of their limitations. The need for validating these models for congenital virology research is reflected by the constant threat posed by emerging viruses. The recent ZIKV outbreak has shown how even a virus that has been circulating in the human population for decades could evolve and cause congenital malformations. Moreover, new human pathogens are discovered each year, some of which could affect human embryonal development. We will discuss the role that RV, as an archetypic teratogenic virus, could play in addressing future research questions on congenital virus infections. The first highlighted aspect is the placenta as an efficient barrier against virus transmission from the mother to the embryo or fetus; the so-called vertical transmission. Thereafter, we will compare the clinical symptoms and epidemiology of congenital rubella to other viruses with teratogenic impact during first trimester infection, namely human cytomegalovirus (HCMV) and ZIKV. Finally, the knowledge obtained from pluripotent stem cell-based cell culture models for the associated developmental steps of human embryogenesis will be discussed. This includes, not only the promises, but also the limitations that are associated with the use of these models.

## 2. All Congenital Viral Infections Face an Efficient Barrier: The Placenta

The main focus of this review is set on defects associated with viral transplacental transmission during the first trimester of pregnancy. Thus, the immunopathological consequences of the maternal immune response to infections and the impairment of placental functions by inflammatory processes are not addressed. For these more indirect aspects of a virus infection, the reader is referred to other reviews, e.g., [3]. Likewise, infections ascending from the urogenital tract are also not discussed. The placenta acts as an efficient barrier against pathogens. Among the genes that contribute to this placental safeguard shield are members of the ISG family, such as IFN λ2 and interferon alpha-inducible protein 6, which are induced in placenta explants upon ZIKV infection [8]. Originally, they were identified as top ranked genes by a clustered regularly interspaced short palindromic repeat (CRISPR) activation screen performed in modified Huh7 hepatocellular carcinoma cells, but were also expressed in ZIKV-infected ex vivo placenta explants [8]. Furthermore, the production of type III IFNs renders trophoblast cells of mid-gestation [9] and full-term placentas [10] refractory to ZIKV infection. This emphasizes the contributory role of the IFN system for the placental barrier function.

The formation of the human placenta is initiated at 5 to 7 days postconception (day 0 is defined as the day when fertilization of the ovum occurs, which is thereafter called the zygote) by the initial contact of the trophoblast cells of the blastocyst with the maternal uterine epithelium, the endometrium, which converts into the decidua. Figure 1A shows the structure of a human pre-implantation blastocyst together with the timeline of the formation of the placenta in relation to the timeline of maternal RV infection that could result in congenital malformations. RV can be teratogenic from early pregnancy onward [11,12], which emphasizes the fact that we need to include the placenta in our considerations and discuss how a virus could infect the developing embryo. The anchorage of the blastocyst is followed by transformation of the trophoblast layer into the syncytiotrophoblast (present as a multinucleated syncytium) and the cytotrophoblast (CTB) as the outer and inner layer, respectively (Figure 1B). At around 5 weeks postconception, the initial CTB cells migrate into the decidua in the direction of the spiral arteries, which initiates vascularization of the placenta [13]. Between 5 and 12 weeks postconception, the maternal circulation is restructured such that the hemochorial human placenta is characterized by direct contact between maternal blood and tissue directly facing the fetal membranes within the placenta [14].

In utero transplacental transmission of pathogens can occur through passage from maternal endothelial microvasculature to endovascular extravillous CTBs or through transfer from infected immune cells to placental syncytiotrophoblasts [13]. Different susceptibilities are prevalent between the two trophoblast cell types, the syncytiotrophoblasts, and the CTBs. Whereas the former is rather resistant to virus infections or shows a low rate of infection, the latter is susceptible to a number of viruses [13]. A detailed and in-depth review on how viruses passage across first-trimester placentas can be found elsewhere [14,15]. In the case of HCMV, the routes of transmission are complex and involve several cell types in the decidua. As shown through HCMV infection of first-trimester decidual organ cultures, CTBs, endothelial cells, macrophages, and dendritic cells were among the cell types targeted, and viral spread occurred preferentially by cell-to-cell transmission [16]. To reach and infect CTBs, immune complexes composed of maternal IgGs and HCMV virions could pass the syncytium through transcytosis based on the neonatal Fc receptor transport system [17]. There is also a wealth of literature available for ZIKV transplacental passage. Multiple studies were performed on mid- and late gestation placentas, explants from first-trimester chorionic villi, human placenta cell lines, human trophoblasts from mid- and late gestation placentas, and even trophoblast cells derived from hESCs, for which a comprehensive overview is provided by Tan et al. in supplementary Tables 1 and 2 [18]. In their study, Tan et al. analyzed ZIKV infection at the stage of pre- and peri-implantation blastocysts as the earliest embryonal stage possible [18]. Blastocyst implantation in the endometrium occurs at a position that allows for attachment of the trophoblast layer to endometrial epithelial cells [19,20]. The invasion of the blastocyst into the epithelium of the uterus results in a multitude of changes of the endometrium, summarized in the term decidualization, and generates an embryonal-maternal interface [19,20]. Through this, the embryonal compartment is potentially accessible for embryotoxic or teratogenic substances or pathogens. Tan et al. used ex vivo mouse and human blastocyst cultures, hESC-derived trophectoderm (TE) cells, and the mouse as an in vivo model to analyze virus susceptibility and infection of pre-and peri-implantation staged blastocysts [18]. Ex vivo infection of mouse and human blastocysts with ZIKV revealed viral replication in TE cells, which was further confirmed by the susceptibility of hESC-derived TE cells to ZIKV. Furthermore, mice were infected with ZIKV at embryonal day (E) 2.5 and E3.5 before implantation of the blastocyst at E4.5. Thereafter, blastocysts were flushed out and cultivated ex vivo for an additional 24 h. The presence of ZIKV-positive cells in these blastocysts confirmed that pre-implantation infection of the dam can be transmitted to pre-implantation embryos [18]. These results challenge our general concept of infections of the developing human embryo. How viruses could get access to pre-implantation blastocysts remains an open question. However, these observations are supported by the clinical data that define the time window during which maternal rubella can cause congenital defects in the embryo (Figure 1A).

In contrast to HCMV and ZIKV, the mode of transplacental passage of RV is rather ill defined and not well studied. Transplacental passage of RV is linked to viremia, as the appearance of the rash in the infected mother can be linked to congenital rubella (Figure 1A). Infection after the last menstrual period and before conception does not lead to congenital defects, whereas the first 11 weeks postconception are critical [11,12]. During viremia, lymphocytes or macrophages can either be infected and support virus replication or harbor the virus [15,21,22], which, as a hypothesis, could enable viral transfer to the hemochorial placenta. While first-trimester trophoblast cells were reported to be resistant to RV infection [23], explants of chorionic villi from human placentas and primary CTBs were susceptible [24]. The study by Töndury et al. was one of the first on a cohort of embryos and fetuses obtained after therapeutic interruption of pregnancies due to maternal rubella during one of the last rubella epidemics [25]. They describe lesions and focal damage of syncytiotrophoblasts and CTBs in the chorion and hypothesize, indicating that the spread of infected, desquamated cells of the chorion to the fetal system could be a possible mode of virus transfer. This is supported by the identification of viral antigens in endothelial cells and basal decidua in a placenta donated by a mother with RV infection in the first trimester of pregnancy [26].

As outlined above, RV can be teratogenic from early pregnancy onwards. Experimental proof is provided for in utero transmission of ZIKV before formation of the placenta is initiated and even before implantation of the blastocyst [18]. Although accessibility of the blastocyst at these early time points of embryogenesis is under discussion, it was already demonstrated in 1959, by Wilson et al., that the teratogenic substance trypan blue can cross the blastocyst wall, resulting in its direct access to the embryo [27]. As a hypothesis, we want to point out the potential role played by maternal immune cells in the transfer of pathogens such as RV in utero as early as one week after conception (Figure 1B). Decidual leukocytes, such as natural killer cells and macrophages, are involved through multiple functions in gestational processes such as decidualization and implantation [3,28]. The trophoblast cells of the blastocyst interact with those immune cells that are infiltrating the decidua, which in turn supports implantation and formation of the placenta. As discussed before, the so far present paradigm of a maternal immune response against a so-called semi-allogenic fetus has changed to a necessary fetal-maternal interaction in support of implantation. This idea of a host-semi allograft model was suggested in 1952 by Sir Peter Medawar and, in analogy to the immune response during organ transplantation, received support through the high proportion of maternal immune cells within the decidua [29,30]. Several studies on the crucial role of these immune cells during pregnancy have revised our view on blastocyst implantation and the development of maternal immune tolerance from a state of immune suppression to a rather dynamic maternal-fetal interface [30]. To name some of the studies, Hanna et al. demonstrated that depletion of natural killer cells within the decidua negatively affects trophoblast invasion [31]. Furthermore, a conditional mouse model of macrophage depletion confirmed the key role of macrophages from early pregnancy onwards, as their depletion during early pregnancy resulted in failure of embryo implantation [32]. As a potentially infected cell population, they could enter the endometrium from circulating blood, providing viral access to the trophoblast cells. The experimental proof for this notion and the identification of ways of in utero transfer around the time point of implantation in general remains a challenge for the future.

## 3. Characteristics of Congenital Virus Infections in Reference to RV as a Representative Teratogenic Virus

RV infection during pregnancy can lead to miscarriage and a wide range of manifestations referred to as congenital rubella syndrome (CRS). As summarized elsewhere [33], the most common defects, as the classic triad, affect the ears (sensorineural deafness), eyes (cataract, glaucoma, retinopathy, and microphthalmia), and heart (patent ductus arteriosus and pulmonary artery). Mental retardation as a sign of central nervous system (CNS) impairment is also noted. While most of the manifestations are permanent, some are of a temporary nature (e.g., hepatosplenomegaly or bone lesions) or become apparent only later in life (e.g., diabetes mellitus type 1). In the first trimester, CRS and, only occasionall, miscarriages and spontaneous abortions occur, but defects are rarely seen after the 16th week of gestation [12,34]. After the initial association of RV infection of the mother and birth defects in the child [1], research efforts centered on potential other viral teratogens. In the mid-seventies, HCMV, herpes simplex virus (HSV), and varizella zoster virus (VZV) had already been identified as likely candidates [35]. While most congenital HCMV infections are asymptomatic, clinical symptoms may include thrombocytopenia, hepatosplenomegaly, intrauterine growth retardation, ocular and auricular disease, and multiple CNS manifestations [36]. After genetic mutations, congenital HCMV infection is the second most common cause of sensorineural hearing loss [37].

For the majority of the aforementioned teratogenic viruses, an essential determinant of the clinical symptoms of the newborn is the gestational age at which infection occurs, e.g., [35,36,38]. Hereafter, we will outline the similarities and differences of the congenital infections caused by HCMV and ZIKV as opposed to RV. Like RV, both viruses can be transmitted to the embryo in the first trimester of pregnancy and cause an analogous congenital syndrome. In contrast to ZIKV and RV, HCMV can not only be vertically transmitted during primary infection, but also after reactivation during pregnancy [39,40]. However, the transmission rate during non-primary or recurrent infection is as low as about 1% [39]. With up to 34% the transmission rate during primary HCMV infection, which is considerably lower than the maximum rate of nearly 90% observed for RV during maternal rubella infection during the first 11 weeks of pregnancy [11,12,39]. While congenital rubella is largely restricted to primary infection during the first trimester of pregnancy, the transmission rate of HCMV during the third trimester of pregnancy occurs at a higher rate than during the first trimester, albeit with less severe symptoms [41]. ZIKV emerged in 2013/2014 in French Polynesia and in 2015 in Brazil [42,43]. The first evidence of perinatal transmission of ZIKV was noted for two infants born in French Polynesia [44]. This report by Besnard on these first two cases was eventually followed by solid proof that ZIKV infection during pregnancy is linked to congenital malformations in the newborn [45,46]. The study by Rasmussen illustratively describes which criteria are necessary to turn an observed coincidence into the identification of ZIKV as a cause of birth defects in humans [45]. Boppana et al. contrast congenital ZIKV to non-ZIKV infections and provide a meaningful comparison of the structural damage caused by HCMV and ZIKV in the brain and spinal cord, as revealed by imaging studies outlined in their review [40]. CNS defects are a less common manifestation of CRS, whereas congenital HCMV and ZIKV infections have a devastating influence on CNS development. This suggests a common pathology of both viruses in relation to the developmental stage of the CNS at the time point of maternal infection [40].

So far, an efficient vaccine is only available for RV, which was implemented in an effective vaccination program and led to a dramatic decrease in the incidence of CRS. However, there are still developing countries in Africa and Asia lacking routine rubella vaccination [33]. Thus, future research efforts on RV are more likely to focus on the elucidation of virus-associated teratogenic mechanisms. As detailed above, the types of symptoms caused are similar for some teratogenic viruses (e.g., neurologic and morphologic symptoms), while some of the most prominent manifestations show variations. Insights into the associated mechanisms including the elucidation of molecular pathways will be highly valuable as benchmark data, not only for the comparative analysis of known teratogens, but also of emerging or even re-emerging viruses with teratogenic potential. This could contribute to the identification of essential cellular markers and antiviral pathways for the conduct of the discovery of curative treatment options. Pluripotent stem cells, especially iPSCs, provide potent tools for the conduction of high-throughput screening (HTS) assays in therapy development, assessment of vaccine safety, and toxicity screenings. Even drugs may represent a toxin when they induce side effects. In the context of ZIKV infection, caspase-3 activity is elevated, representing a promising target for HTS [47]. HTS of compound collections in human neural progenitors, astrocytes, and organoids revealed small molecules, such as the pan-caspase inhibitor Emricasan, as neuroprotective and antiviral agents [48]. Today, computational design, in vitro cell culture applications, in vivo animal models, and clinical trials are part of the pipeline. Structural-bioinformatics-based methodologies and frameworks are suitable for the design of ZIKV monoclonal antibodies against five envelope proteins and to estimate their performance in vitro or in vivo [49]. These candidate compounds need to be transferred to appropriate HTS assays and require appropriate cell culture models. The application of iPSCs may represent a promising advance to improve safety prediction, especially when patients are pregnant women.

## 4. Implications of Genetic Variations of Teratogenic Viruses

Viral pathogenesis and disease-associated characteristics, such as those occurring during congenital infections, can differ among clinical isolates of human-pathogenic viruses. Such strain-specific differences occur in vitro as well as in vivo through viral adaptation to its host cell or human host, respectively. In the case of HCMV, these adaptations can occur at a fast rate. A recent review summarizes how this could hamper research on HCMV. Low-passaged HCMV strains are usually cell-associated, which restricts their use in experimental characterizations. During cell culture adaptation, HCMV strains acquire mutations and a reduced level of cell association and, thus, produce higher titers [50]. Moreover, some adaptations appear to be dependent on the cell type used for its passage. These in vitro adaptations during cultivation of HCMV in cell culture are complemented by the in vivo variation among HCMV isolates, which reflects their complex interaction with the host immune system [50]. Within the human population, viruses are confronted with a high degree of variation of their host’s antiviral immune response mechanisms [50]. Direct sequencing of as much as 91 HCMV strains from clinical material indicated that distribution and percentage of mutations within viral genes were comparable to those noted for cell culture-passaged strains [51]. Moreover, sequencing of samples isolated from congenitally infected infants revealed that some mutations were associated at a higher frequency with symptomatic infections [52].

The two ZIKV lineages, Asian and African, display differences in their associated pathology and, especially infections with the Asian lineage, may result in severe malformations and developmental defects [43]. However, after in utero inoculation of ZIKV in a fetal pig model, the trans-fetal ZIKV transmission and the viral loads in the placenta, as well as cerebrum and cerebellum of the fetuses, were higher for African than for Asian strains [53]. This emphasizes that further studies on the epidemiology and the heterogeneity of ZIKV strains are required [53]. Genomic epidemiology contributes to our understanding of how changes within the viral genome could contribute to changes in viral pathology. One amino acid substitution within the precursor membrane protein of ZIKV was sufficient to result in more microcephaly in a mouse embryonic microcephaly model and in a higher rate of infectivity and cell death induction on mouse and human neural progenitor cells (NPCs) as compared to the parental virus [54]. Furthermore, among ZIKV strains, differences in the capacity to induce an antiviral innate immune response and sensitivity to IFN were detected [55]. The innate immune evasion strategy, especially displayed by the emerging Brazilian strains, could contribute to viral pathogenesis during congenital infection [55]. In contrast to HCMV and ZIKV, RV displays only a low level of variation within its low-passaged clinical isolates. While differences in the adaptation of cellular metabolism and the requirement for glutamine were detected [56], a comparable level of innate immune activation was identified [22]. Moreover, the number of circulating RV strains has decreased in the last decade. The most recent WHO RV nomenclature update in 2013 proposed one provisional (1a) and 12 recognized (1B–1J and 2A–2C) genotypes belonging to two clades (1 and 2), [57]. In 2018, strains belonging to only two genotypes (1E and 2B) were reported to the global Rubella Nucleotide Surveillance (RubeNS) database (www.who-rubella.org/files/wer8832.pdf), [58]. In many countries, genotype replacements were noted (e.g., [59,60,61]) and lately involve, most often, genotype 2B as the new dominant genotype. While the reasons for the observed dominance are unknown, the question has been raised whether 2B strains have higher transmission ability [62]. This epidemiological change towards one dominating genotype, in addition to the presence of just one serotype for RV, can be considered advantageous for its use as an archetypic congenital virus model. Genetic variants present within RV genotypes are less likely to influence congenital pathogenicity. Thus, only the currently dominating genotype 2B appears to be sufficient for the assessment of RV on cell culture models for human embryonal development.

## 5. Experimental Animal Models and the knowns and unknowns of Viral Teratogenicity in Humans

The application of experimental animal models for congenital infections in humans reveals essential disease mechanisms, but comes with several restrictions due to human-specific aspects. In the case of HCMV, the newborn mouse and guinea pig model were employed in assessment of virus-induced brain pathology [63] and vaccine evaluation [64], respectively. However, species specificity in general limits the use of animal models to the analysis of only some of the symptoms of congenital HCMV infection [16,65]. Mouse cytomegalovirus (MCMV) was used as a close relative of HCMV in the mouse model and, as such, as a stand in for congenital HCMV infections. This includes intraperitoneal inoculation of newborn mice with MCMV as a neurodevelopmental counterpart of second trimester human fetuses [65]. In this model, no signs of direct cytopathology in the inner ear were detected, but loss of neurons in the spiral ganglia and an inflammatory response in the cochlea were identified [65]. This corresponds to the delayed onset of hearing deficits or their progression over time in children with sensorineural hearing defects [66,67]. Furthermore, intraplacentally inoculated MCMV revealed impairment of the olfactory bulb structure and a reduced number of bulbar dopaminergic cells, together with an olfactory deficit as a sign of neuronal damage that can be noted before the onset of hearing loss [68].

Comparable to other viruses that are transmitted by arthropod vectors, IFNAR−/− mice lacking the receptor for type I interferons (IFN-α and -β) are a suitable experimental model for ZIKV [69]. Any type I interferon that is produced by infected cells within this animal model is not bound by their receptor, neither on the infected cell itself (autocrine) nor on neighboring, uninfected cells (paracrine). Thus, the type I IFN-associated antiviral state as a crucial determinant of susceptibility to ZIKV is not generated. Immunocompetent wild-type mice are resistant to ZIKV and lack disease symptoms and signs of virus replication [70]. In humans, ZIKV counteracts IFN pathway activation through targeting signal transducer of transcription 2 (STAT2) for degradation by the viral non-structural protein 5 [71]. This viral strategy of immune evasion is not active against mouse STAT2 [71]. Tan et al. provide, in their study on ex vivo and in vivo ZIKV infection in mice, a detailed summary of the literature on the IFNAR−/− mouse model and the use of anti-IFNAR1 blocking antibodies along with ZIKV infection [18], (Supplementary Material, Table 2 of [18]). Eventually, this fundamental finding led to the generation of a human STAT2-knock-in immunocompetent mouse model for ZIKV [72]. Pregnant mouse models of ZIKV infection provided valuable insights into congenital ZIKV infection through recapitulation of several aspects identified in the human disease [73]. Non-human primates are a more suitable model for human pregnancy than mice, which is related to the structure of their placenta and the gestational time, but depends on the species (pigtail and rhesus macaques) used as a model for congenital ZIKV infection [74]. However, in comparison to humans, these non-human primate models appear to display a prolonged viremia [73]. Additionally, rather, pathologic consequences are noted in association with the virus-induced immune response. Thus, the validation of their suitability as animal models for congenital ZIKV infection necessitates further research [73].

Although transplacental passage of RV was demonstrated for some animal species, including rhesus monkeys [75,76], none of them reflected the persistent infection in human fetuses, or the teratogenic effects noted for congenital RV infection. In the study published by Parkman in 1965, pregnant rhesus monkeys were inoculated intravenously or intramuscularly with RV during the 4th or 19th week of gestation. However, although several aspects of the human course of infection, such as rate of transmission to the fetus and the presence of a viremia in pregnant animals, were reflected, no malformations were present in the embryos and fetuses [75]. This complements the restrictions of the use of animal models in congenital virology as outlined before, which includes the use of MCMV as mouse homolog for HCMV and, in the case of ZIKV, the requirement of an immunocompromised mouse model for disease pathology assessment.

## 6. Novel Cell Culture Models to Address the unknown Mechanisms of Virus-Associated Alterations

### 6.1. The Advent of Induced Pluripotent Stem Cells and the Associated Implications for Developmental Biology

While the mouse model in general has provided valuable information for human development and, more specifically as outlined above, on congenital ZIKV syndrome (CZS) and the associated pathology, it faces important restrictions. The interspecies differences between human and animal development and the awareness of the three Rs (3Rs) principle for animal research (Replace, Reduce, Refine) ask for alternative research approaches. On the one hand, humans and mice share a hemochorial placenta, 99% of their genes, and most of their protein-coding genes [77,78]. On the other hand, just to name a few differences, brain and eye development in humans are more complex than in mice [79] and notable discrepancies exist in gene expression patterns during development [80]. Correspondingly, some mutations and knock-outs of genes in mice resemble the phenotype observed in humans, and others do not [77]. Even for pre-implantation staged blastocysts, considerable differences exist between humans and mice. The knock-out of the pluripotency transcription factor octamer binding transcription factor 4 (OCT4) in human embryos by CRISPR/Cas9 gene editing technology prevented their development to the blastocyst stage, whereas mouse embryos were stopped at a later stage [81]. These arguments emphasize the need for alternative ways to study human development. The closest view we probably could get on the earliest stages of human development was made possible through research achievements published recently by two groups; the groups of Zernicka-Goetz and of Ali Brivanlou, on the in vitro cultivation of human embryos donated by women after conducting in vitro fertilization programs [82,83]. They reported on in vitro cultivation of human blastocysts for an extended period of time in agreement with the ethical constraints posed by the so-called 14-day rule. Some European countries, such as the United Kingdom, Belgium, and the Netherlands allow research on human embryos until day 14 of human development, including the use of ESCs. As an additional resource for conducting research on human development, sectioned human embryonic or fetal tissue samples are available through tissue banks, such as the Human Developmental Biology Resource (HDBR) in the UK [79]. However, the embryonic and fetal tissues provided by HDBR are restricted to 3 to 20 weeks of development (www.hdbr.org). Moreover, for technical and ethical reasons, the use of embryos and the samples provided by these tissue banks is fairly limited. Nevertheless, they could be used to verify findings on pluripotent stem cells (embryonal and induced). ESCs and iPSCs are especially a cell culture platform with unprecedented possibilities to address human development. Thus, a highly promising approach based on pluripotent stem cells is, not only moving into the focus of developmental biology, but also of congenital virology. Figure 2A highlights, not only the potentials, but also the limitations of the experimental model systems that are currently available to study the impact of congenital virus infections on human development.

As highlighted in Figure 2A, human population-based studies can identify host genetic factors with an impact on susceptibility to a virus infection or severity of the associated disease. The challenge in the field of congenital virus infections lies in the differential host genetic contribution by the mother and the fetus or newborn on disease progression as their genetic background is not identical. Additionally, the immune response appears to be of central importance, which could mask genetic variants of developmental pathway components. One of the examples of contributing immunological factors is toll-like receptor 2 (TLR2) 2258 G>A single nucleotide polymorphism (SNP), which was associated with a higher risk of congenital CMV infection in a cohort of Polish fetuses and newborns [84]. Host genetics is a rather new approach in congenital virology and, especially during the recent ZIKV epidemics in Brazil, host genetic factors were identified. A case-control study of a Brazilian cohort identified a statistically significant association of maternal adenylate cyclase genes with the development of CZS [85]. Comparable to the identification of the TLR2 2258 G>A SNP, the T allele in SNP rs3775291 of the TLR3 gene is associated with a risk of CZS [86]. Furthermore, a link between the T allele in SNP rs1799964 at tumor necrosis factor α and the severity of ZIKV-induced microcephaly was identified [86]. The potential these case-control studies hold for hiPSC-based models was revealed by the analysis of possible host genetic variants in NPCs derived from hiPSC generated from erythroblasts from a healthy control group and from the following pairs of Brazilian twins that were exposed to ZIKV during pregnancy: monozygotic twins equally affected (concordant) and dizygotic twins with one CZS-affected and one unaffected (discordant) twin [87]. RNA sequencing (RNA-Seq) performed on these NPCs identified DDIT4L as the gene with the most significant difference in expression between the CZS-affected and the unaffected group. The reduction of DDIT4L expression as an inhibitor of mammalian target of rapamycin mTOR signaling was accompanied by differences in the activity of the mTOR pathway between these two groups [87]. Although no main gene locus was assigned, individual differences in host signaling pathways such as mTOR reflect differences in the host response to a virus infection [87]. On the one hand, iPSCs carrying a certain DNA risk variant provide a powerful tool for the functional analysis of DNA variants and their impact on onset, progression, and treatment of infectious diseases. On the other hand, in combination with genetic engineering approaches, they could compensate for the lack of healthy controls with the identical genetic background during population-based studies (Figure 2A).

Pluripotent stem cell-based models mimic early human development through the application of various defined protocols to initiate differentiation. This can be done in two or in three dimensions, as exemplified by the formation of stem cell aggregates, the so-called embryoid bodies (EBs). In vitro differentiation can be spontaneous (undirected), as in the case of EB formation, or directed along a specific developmental route. Directed differentiation can also be initiated in adherent cell cultures or EBs through the addition of defined growth factors and pharmacological compounds [77,88]. Lineage commitment of pluripotent stem cells during directed differentiation into the three embryonic germ layers, ecto-, meso-, and endoderm, closely reflects signaling events and networks during embryogenesis. Figure 2B illustrates the first steps of human embryogenesis, which can be modelled with pluripotent stem cells. There is continuous research on the reproduction of spatial processes within such model systems and culminates so far in the morphogenesis of organoids such as the optic cup [89,90]. In addition to these retinal organoids, the ability to reflect the spatial organization of the embryo in vitro is also shown through cerebral organoids as three-dimensional, multicellular models for studying the processes during organogenesis [90]. Besides these steps that follow implantation of the blastocyst, a recent addition to the cabinet filled with technological breakthroughs in stem cell biology is the three-channel microfluidic device with matrix pouches as a hESCs- or hiPSCs-based advanced model of the spatial events at the peri-implantation stage [91]. Within this continuously advancing field, research questions on congenital virus infections are just beginning to be addressed, which we will outline hereafter.

### 6.2. The infection of Pluripotent Stem Cell-Based Models of Human Embryogenesis with Teratogenic Viruses

In the context of the implications that pluripotent stem cell-based models may have for our current concept of congenital virology, a justified question may arise on their relevance for addressing research questions on congenital infections. Human embryonic stem cell lines originate from the inner cell mass of human blastocysts, which were derived through cultivation of embryos donated by participants of in vitro fertilization programs [92]. The application of ZIKV in the IFNAR^-/-^ mouse model confirmed experimentally in the vertical transmission mouse model that pre-implantation infection of the dam can be transmitted to pre-implantation blastocysts [18]. This emphasizes the relevance of ESCs and, as such, iPSCs for the study of viral alterations during the very early stages of human pregnancies. The study by Tan et al. also revealed a dose-dependent effect of ZIKV on blastocyst proliferation, which complied with the paradigm shift of the all-or-none hypothesis, as outlined before. At high viral doses of ZIKV, infection resulted in cell death of TE cells, whereas low doses allowed for viral propagation and potentially for in vivo transmission of infectious viruses to other cell populations during embryonal development [18].

The implication of the two pluripotent stem cell lines, ESCs and iPSCs, as a suitable model system for congenital virology is two-fold. First, the outer-lining of the blastocyst, the trophoblast, can be modelled through trophoblast differentiation of pluripotent stem cells. Second, the epiblast, which gives rise to the three embryonic germ layers, can be recapitulated through two- and three-dimensional differentiation approaches and reflect early gastrulation-like developmental stages. Already in 2002, the first protocol on differentiation of hESCs into trophoblasts was published [93]. While the addition of BMP4 to the culture medium of ESCs induces mesoderm formation, TE cells are induced in the presence of BMP4 besides feeder-conditioned media [93,94]. Li et al. could show that hESCs differentiate into an early CTB-like phenotype before further differentiation into terminally differentiated syncytiotrophoblasts and extravillous trophoblasts. A similar approach of TE differentiation of ESCs was pursued by Tan et al., which generated hESC-derived TE cells and confirmed that they were identical to the in vivo counterparts isolated from human blastocysts [18]. These ESC-derived TE cells are susceptible to ZIKV, with a dose-dependent virus infection and propagation rate [18].

Several studies were conducted on the impact of virus infections on mouse as well as on human pluripotent stem cells. Here, we want to focus on the findings related to the impact of HCMV, ZIKV, and RV on human development. Early reports on embryonal carcinoma cells suggested that HCMV permissiveness could change during the transition of the undifferentiated (pluripotent) to the differentiated state [95]. The susceptibility of these cells was very low and possibly restricted to cells undergoing spontaneous differentiation. Upon treatment with retinoic acid, not only differentiation was induced, but susceptibility to HCMV also increased [95]. This block in replication, which appeared to occur at steps following entry, was also reflected through later studies on the assessment of the suitability of different promoter types for transgene expression in pluripotent stem cells [95,96,97,98]. This did not only support the use of the elongation factor-1 alpha promoter for genetic engineering of human ESC and iPSCs [97,99], but also the notion on the relevance of such differences in epigenetic regulation between pluripotent stem cells and somatic cells for replication of HCMV. For MCMV, as a relative of HCMV, non-permissiveness of mouse ESCs was demonstrated, in addition to a lack of activity of the MCMV IE promoter in ESCs derived from MCMV immediate-early (IE) promoter-lacZ transgenic mice [100]. Moreover, IE promoter activity and correlatively permissiveness were dependent on the state of differentiation, and especially present in glial differentiated cells [100]. As a follow-up, two studies on the effect of pluripotency on permissiveness to HCMV were published in 2012 and 2014 [101,102]. Both studies highlighted the nonpermissiveness of iPSCs to HCMV, whereas NPCs were susceptible and fully permissive. Furthermore, the study by Belzile et al. identified different restrictions in ESC-derived primitive prerosette neural progenitor cells (pNPC), including one at the level of IE gene expression [101]. This block was associated with the presence of the neuronal developmental marker FORSE-1 and was partially lost after pre-treatment of pNPCs with retinoic acid and further differentiation into NPCs. Despite this inhibition in replication, viral genomes persisted in pNPCs [101]. Due to their similarity to cells in the developing neural tube, they are suggestive of the presence of a cellular harbor at a very primitive neuronal developmental stage for virus persistence along further neuronal differentiation stages [101]. This notion of a gradual release of viral restriction blocks along neuronal differentiation was corroborated by the transition to an increased susceptibility to HCMV during neural differentiation of ESCs into neural spheres [103]. Susceptibility to HCMV appeared to be correlated to expression of the platelet-derived growth factor receptor alpha (PDGFRα) [103]. Brown et al. showed correspondingly a lack of permissiveness of iPSCs to HCMV, but also identified iPSCs to be susceptible to HCMV infection through positive staining for the input tegument proteins [104]. Brown et al. utilized such observations for the generation of cerebral organoids through differentiation of infected iPSCs to avoid experimental challenges for infection based on the compact three-dimensional structure of the organoid [104]. These cerebral organoids are promising as a model for congenital HCMV infection and its impact on the early stages of neuronal development. The following findings are correlated to the scattered distribution of viral antigen-positive cells in cortical sections of clinical tissue samples and the cortical lesions found in those samples; besides signs of necrosis and the presence of vacuoles and cysts in cortical structures, HCMV infection affected the patterning of postmitotic, differentiated neurons and reduced the number of true cortical structures [104,105]. Thus, this reflects a promising approach to identify cellular mechanisms contributing to the severe effects of congenital HCMV infection on neuronal development [40].

In contrast to infectivity by HCMV, iPSCs are susceptible and permissive to ZIKV and RV [47,106]. Among the plethora of original research articles and review literature on pluripotent stem cell-derived models for ZIKV-associated microcephaly and its neurotropism, we can only highlight a few. The models summarized hereafter [107,108] reflect developmental stages of 5 to 8 up to 10-week-old human embryos. Defined chemical conditions allowed for differentiation of pluripotent stem cells into NPCs from forebrain dorsal, forebrain ventral, and hindbrain and spinal cord brain organoids. Their infection with ZIKV was associated with growth arrest and apoptosis. Additionally, overactivation of ISGs was induced in an IFN-independent manner through IRF3- and NF-ΚB-associated pathways [107]. The study by Dang et al. used immature three-dimensonal cerebral organoids after 10 days of differentiation as a model for the impact of ZIKV on neurodevelopmental processes. After five days of infection, a reduction in organoid size was detected that appeared to be due to loss of the NPC pool as the main cell type targeted by ZIKV. Additionally, activation of TLR3 affected TLR3-regulated networks that direct neurogenesis and apoptosis within NPCs of these cerebral organoids [108]. This is in contrast to the findings of Liu et al., which did not identify boosting of TLR3 signaling during ZIKV infection [107]. This may be due to the different NPC populations involved in both studies; rather pure NPCs by Liu et al. as opposed to NPCs within cerebral organoids of a later stage of differentiation by Dang et al. [107,108]. This highlights how important it is to consider, not only different cell types, but also stage-specific effects within these models while addressing mechanisms during congenital virus infections. Complementing this data on the impact of ZIKV on cerebral organoids, iPSC-derived human retinal epithelium (RPE) was employed to address ocular defects noted after congenital ZIKV infection [109]. Transepithelial resistance of the RPE, as well as cell junctions and, thus, RPE architecture, were impaired after ZIKV infection. As a follow-up of these observations, a strong inflammatory response, as revealed by induction of IFNs, was identified in RPE after ZIKV infection [110]. The relevance of inflammation as an effective host countermeasure against ZIKV infection is emphasized by the sex determining region Y (SRY)-box 2 (SOX2)-dependent susceptibility of glioblastoma stem cells, as SOX2 expression was associated with a reduced innate antiviral response [111].

The unique ability of RV to replicate in iPSCs without cytopathogenic alterations enabled passaging of RV-infected iPSCs and their differentiation into embryonic germ layer cells [106]. However, their aggregation into EBs was impaired, which appeared to be due to a reduced level of expression of genes involved in cell adhesion. RV infection does not possess the capacity to direct differentiation of iPSCs into a distinct developmental pathway as the transcriptomic profile of mock-infected iPSCs after induction of undirected differentiation was comparable to those infected with RV [106]. Directed differentiation of RV-infected iPSCs into embryonic germ layer cells revealed dysregulated developmental pathways, such as Wnt in mesodermal cells. Specifically, compared to the uninfected control, a downregulation of eye field transcription factors such as Six homeobox 3 (SIX3) and retina and anterior neural fold homeobox (RAX) involved in eye development was found in RV-infected ectodermal cells [106]. Moreover, endodermal differentiation especially, was affected as indicated through an increased expression of markers for definitive endoderm such as Eomesodermin (EOMES) and Cerberus (CER). Through cooperative crosstalk with the precardiac mesoderm, the endoderm is involved in specification of cells of the cardiac lineage [112]. These findings in ectodermal cells on dysregulation of genes involved in eye development and the upregulation of definitive endoderm markers, together with an altered Wnt activity in mesodermal cells, can be correlated to retinal and cardiac defects, respectively. These are two of the main CRS symptoms. These examples highlight the value of iPSC/ESC-based cell culture models for the study of congenital virus infections. During congenital infection, all three viruses, HCMV, ZIKV, and RV can cause eye defects. The analysis of their course of infection during ectodermal differentiation, and more specifically on retinal organoids, could identify common, but also different, strategies pursued by the three viruses. These strategies need to be identified and compared to get a complete picture of the mechanisms of congenital virus infections.

### 6.3. Thoughts on the Limitations and Promises of the Continuously Advancing Array of Pluripotent Stem Cell-Based Models

Human ESCs and iPSCs share almost all characteristics of pluripotent cells. However, gene expression profiles of different iPS cell lines are very similar, but not identical to human ESCs [113]. Some issues remain due to differently expressed genes that are retained from the somatic donor cells or are induced during reprogramming. Moreover, different cell types are used for generation of iPSCs in combination with a diversity of reprogramming procedures, as addressed in a recent review [113]. This leads to variations in reprogramming efficiencies and iPSC quality and, subsequently, to a heterogeneity of iPS cell lines as a major challenge of iPSC-based research questions.

Pluripotent stem cell-based in vitro models grow increasingly complex. Moreover, the current technological advancements in the research on early human embryogenesis offer great possibilities for the understanding of diseases and for finding new therapies. Organoid cultures and engineered tissue cultures might reach a complexity that is comparable to the one found in tissues and organs of the adult human body. As a technical advance in the field, three-dimensional bioprintings of human alveolar cells were infected with the seasonal influenza virus to mimic the clustered infection pattern that is also observed in the lung of human beings [114]. Accordingly, the advancement of our research questions requires ethical adjustments, legal regulations, and definitions in the context of scientific use and clinical application, such as transplantations. This applies especially to the generation of embryo-like structures that need to respect legal limitations and ethical constraints including the 14-day rule [115]. One of the important future challenges is the way we will address these ethical questions.

Currently, rodents are broadly applied for the modelling of infectious diseases including congenital viral infections. One can conclude that species-specific differences most probably represent a major road block for the development of drugs and vaccines besides toxicological testing. However, species differences could also broaden our view on the evolution of humans and the co-evolution of congenital virus infections. Through the application of the iPSC technology to various species, including members of the great apes, we could address species-specific differences in congenital virus infections. Similar studies have already been conducted to address questions on human evolution. The comparison of endodermal differentiation that was induced in human and chimpanzee iPSCs revealed conserved, but also divergent evolutionary circuits [116]. In turn, recent work by the group of Michael Drukker indicates that the findings on gene-regulatory networks in human pluripotent stem cells could be verified in vivo using monkey embryos. In their study, they have not only confirmed the essential role played by GATA3 during TE specification through in vitro assays based on human pluripotent stem cells, but they have also assessed the in vivo role of GATA3 in embryonic development in primates through microinjection of GATA3 morpholino antisense oligonucleotides into zygotes of rhesus macaques. Zygotes were differentiated in vitro and analyzed, which revealed embryonal arrest at the 32-cell morula state [117]. This approach combines stem cell technology with classical animal models in a so far unprecedented way.

Pluripotent stem cells enable the differentiation of mature and functional cells, tissues, and organoids mimicking the (1) differentiation and maturation status of cells, (2) the specific cell composition of a certain tissue, (3) physical impulses necessary for maturation, and (4) the three-dimensional structure of certain tissues. However, there are technical limitations for reconstructing embryonal and developing tissues and organs for a more precise analysis of congenital virus infections. Accordingly, there is a need for pluripotent stem cell-based models for advanced three-dimensional in vitro models that closely resemble the in vivo situation. The importance of such three-dimensional scaffolds is exemplified by brain organoids that, in contrast to neuronal cell culture, reflect at least some parts of the architecture in early embryonal brain tissue. The recent review by Chen et al. does not only summarize the advancements in brain organoid technology over the past five years, but also their application in the study of human diseases, including ZIKV microcephaly [118]. This review by Chen et al. also covers recent drug screening approaches against ZIKV on brain organoids, which allow for assessment of antiviral drugs on brain organoid structure and viability and integrity of the cortex. Among those is 25-hydroxycholesterol (25HC), which is converted from cholesterol by the enzyme cholesterol 25-hydroxylase and for which an antiviral effectiveness against ZIKV was shown in animal studies, whereas the reduction of ZIKV-induced cell death was not appreciable on human brain organoids [119]. This initiated further testing of antimicrobial compounds, including antibiotics, and identified duramycin and ivermectin as promising drugs [119]. Furthermore, brain organoids allow for genetic engineering assays, which will provide mechanistic insights into the ways neurotropic viruses such as ZIKV interact with their human host. They also complement vaccine development and the assessment of vaccine efficacy and safety. Together, two-dimensional and three-dimensional pluripotent stem cell-based in vitro models hold great promise for the understanding of congenital viral diseases and the currently poorly understood infection of the embryo. However, technical advancement is necessary in the field of translational medicine aimed at the transfer of stem cell-based virology research from bench to bedside.

## 7. Conclusions

With this review, we have addressed the possibilities offered by novel stem cell-based cell culture models for the study of the clinical presentation following congenital virus infections. We also want to stress the potential use of viruses such as HCMV, ZIKV, and RV, with their well-defined clinical symptoms during congenital virus infections, for the identification of novel gene functions and regulatory networks during human embryonal development. The combination of congenital virology with the continuous advancements of stem cell-based technology will complement developmental biology and extend our current way of getting information on the function of genes that are active in human development. Our knowledge of human development is, so far, to a large extent fueled by our knowledge from naturally occurring mutations. Data gained on congenital virus infections in stem cell-based models will also help us to better understand deleterious pregnancy outcomes. Moreover, the validation of iPSC-based cell culture models for the study of human development through infection with teratogenic viruses with well-known aetiologies will provide a framework for pathogens with unknown teratogenic potential. We will be able to meet the constraints posed by emerging viruses and to confirm the safety of viral vaccines for their application during pregnancies. We have never been so close to the experimental assessment of the open questions from teratogenic alterations during human pregnancies. Despite the remaining challenges, this is an exciting outlook.

## Figures and Tables

**Figure 1 cells-09-00542-f001:**
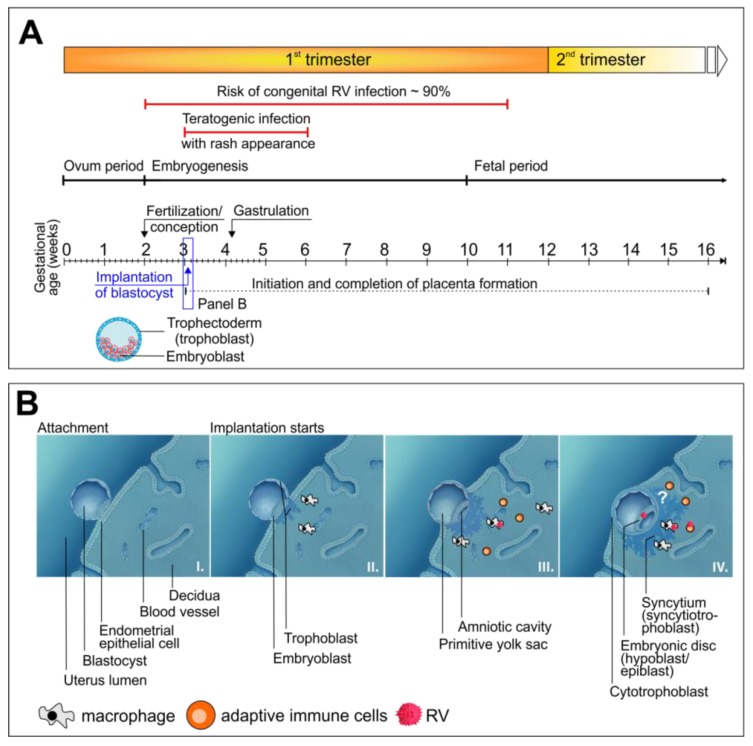
Timeline of congenital defects from maternal rubella virus (RV) infection in relation to embryonal stages. (**A**) The timeline of maternal RV infection with a high rate of congenital defects is given in direct comparison to first trimester embryonal development. The timeline refers to gestational age and starts from the first day of the last menstrual period. Highlighted is the blastocyst, which is composed of the embryoblast as the inner cell mass and the trophectoderm or trophoblast as its outer cell layer. (**B**) The inner cell mass of the blastocyst, the so-called embryoblast, transforms into a double layer, the hypoblast and epiblast. The hypoblast or primitive endoderm develops into the fluid-filled amniotic cavity, which later harbors the embryo. The epiblast gives rise to the embryo, which results in formation of the embryonic disc. The implantation of the blastocyst is followed by infiltration with maternal immune cells, which could potentially provide access of RV to the embryo. The question mark indicates this hypothesized mode of viral transfer. As a reference for (**A**) and (**B**), please refer to the main text, especially for the time line of congenital rubella infection in relation to gestational age to [11,14]. The timeline for teratogenic RV infection in relation to appearance of the rash in the mother is given in [12].

**Figure 2 cells-09-00542-f002:**
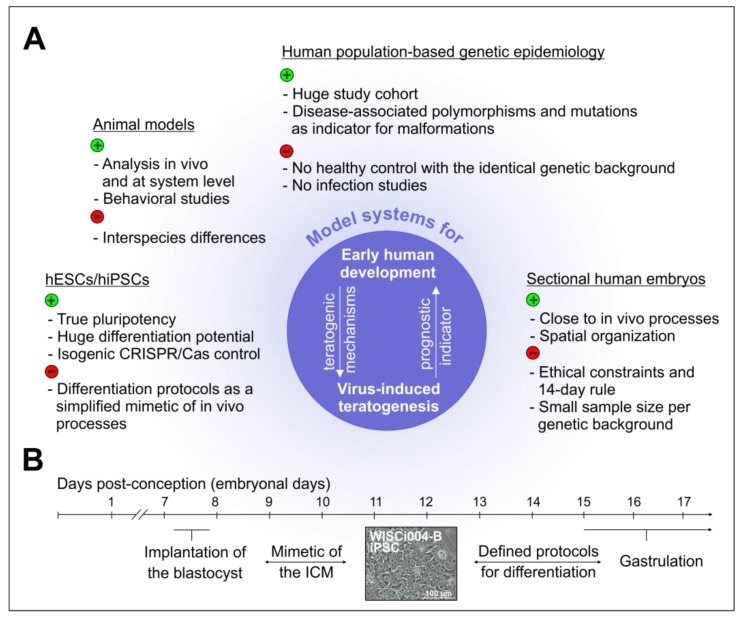
Illustrative summary of current models available for the study of congenital virus infections. (**A**) Contrasting juxtaposition of the respective advantages and disadvantages of in vitro and vivo models for human development. (**B**) Timeline for the very early steps of human embryonal development that are addressable by undifferentiated and differentiating pluripotent stem cells as representatively illustrated by WISCi004-B iPSCs. Timeline is given in embryonal days as compared to gestational age in Figure 1A. As a reference, the reader is referred to the main manuscript text.

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
