# Peer review of "Pluripotent Stem Cell-Based Models: A Peephole into Virus Infections during Early Pregnancy"

_cells, 2020, doi:10.3390/cells9030542_

Round 1

Reviewer 1 Report

The manuscript "The old foe rubella virus joins cytomegalovirus and Zika virus in novel models for congenital virus infections" reviews past and present models for studying virus-induced congenital disease. The basic premise of the review is sound and the authors include a wealth of information on previous models for investigating congenital disease and current trends for using pluripotent stem cells, including their limitations. The wealth of detail, and the extensive use of acronyms, makes this in places a tough read and reducing the length may help make this a more concise and digestible review. Even the title could better reflect the content of the review, as it implies that the viruses are the models. The following comments are a mix of general and specific comments:

Line 88, assume this should read ".. of early embryonal development remains ill-defined." to make sense of this statement. Line 232, "..to as congentical rubella syndrome (CRS)." Line 244, "..overt ocular"? Line 286, "..for the conduct of .." Line 427, This paragraph doesn't really say anything. What does "one certain donor" mean? The sentence "The occurance ... [88], Line 438" should be followed by examples of the DNA variants. Line 454, What are "behind the scenes questions"? Line 499, The paragraph provides too much detail and could be removed. The inclusion of a sentence on ZIKV seems like an after-thought. Line 577, "In contrast to infection by HCMV.." Line 618, give evidence for this statement on correlation. Line 698, "clinical presentation following congenital" Line 701, "and extend our" Line 703, "a large extent"

Author Response

General comments and general changes made to the original manuscript

The authors want to thank the reviewers for their time spent on our manuscript. Their helpful comments and suggestions for revision improved the manuscript.

Point-by-point response to the reviewer’s suggestions

Reviewer 1

The basic premise of the review is sound and the authors include a wealth of information on previous models for investigating congenital disease and current trends for using pluripotent stem cells, including their limitations.

# Thank you for this positive feedback.

The wealth of detail, and the extensive use of acronyms, makes this in places a tough read and reducing the length may help make this a more concise and digestible review.

# We have reduced the length of the manuscript by two pages.

Even the title could better reflect the content of the review, as it implies that the viruses are the models.

# Title was changed to “Pluripotent stem cell-based models: a peephole into virus infections during early pregnancy”

Line 88, assume this should read ".. of early embryonal development remains ill-defined." to make sense of this statement.

# The sentence was revised as follows: Transplacental transmission, the crosstalk at the embryo/fetal-maternal compartment and the mode of impairment of early embryonal development remain not only for RV ill-defined.

Line 232, "..to as congentical rubella syndrome (CRS)."

# The abbreviation was already introduced in line 211. We have revised the sentence in 211, such that the term CRS is now introduced in line 268 in the context of the disease.

Line 244, "..overt ocular"?

# The sentence was revised as follows: While most congenital HCMV infections are asymptomatic clinical symptoms may include thrombocytopenia, hepatosplenomegaly, intrauterine growth retardation, ocular and auricular disease and multiple CNS manifestations.

Line 286, "..for the conduct of .."

# Changed as suggested.

Line 427, This paragraph doesn't really say anything. What does "one certain donor" mean? The sentence "The occurance ... [88], Line 438" should be followed by examples of the DNA variants.

# The paragraph was revised and we have added several examples of DNA variants that are associated with congenital virus infections (line 533 to line 566).

Line 454, What are "behind the scenes questions"?

# This information is not required for this paragraph and was thus deleted.

Line 499, The paragraph provides too much detail and could be removed. The inclusion of a sentence on ZIKV seems like an after-thought.

# We have shortened this paragraph, such that the relevant notion on accessibility of pre- and peri-implantation staged blastocysts is emphasized. The authors consider this as an important notion as it highlights that iPSCs are a relevant cell-type for developmental stages during early pregnancy as they are potentially accessible to virus infections of the mother during pregnancy.

Line 577, "In contrast to infection by HCMV.."

# Changed as suggested.

Line 618, give evidence for this statement on correlation.

# The paragraph was revised to clarify the evidence for this statement.

Line 698, "clinical presentation following congenital"

# Changed as suggested.

Line 701, "and extend our"

# Changed as suggested.

Line 703, "a large extent"

# Changed as suggested.

Reviewer 2 Report

The authors present an overview of recent developments in teratogenic viruses (RV, HCMV, ZIKV), with a strong focus on stem cell models of infection. While generally well-written and presented, with interesting insights and descriptions of cellular and organoid models for these infections, there are a few issues I have outlined below pertaining to the clarity of the text and the presentation of adequate primary literature for some of the sections. Furthermore, I have a major concern with the inclusion of unpublished, non-peer reviewed data (line 544-546, Figure 3) from the authors themselves in the manuscript. This should not be present in a balanced review article of the published literature in the field, and should be removed in the final version of the paper.

General comment: The authors frequently refer the reader to external reviews, usually in a passing sentence, which is not great form for a review paper. While this is OK to do a few times, it is done too frequently in this paper (there are more instances than the two listed in the specific comments). Either delete some of these sections, or cite the primary literature and expand on these ideas.

Specific comments:

Line 32: should be ": it [...]"

Line 34: no comma

Line 67-68: reorganize the text to have a discussion of the role of interferons in viral pathogenesis before the discussion of IFNAR knockout mice (to provide a logical flow for the non-expert)

Line 84: "advance. Teratogenic [...]" should be "advance, but teratogenic [...]"

Line 87: rephrase sentence so that it is not a fragment

Line 92: "un-accessible" should be "inaccessible"

Line 113: rephrase: "The placenta acts as an efficient barrier against pathogens."

Line 115: comma after "family", "which were" should be "which are"

Line 128: "on" should be "onward"

Line 176: comma after "viremia"

Line 233: define "CRS" acronym in the text here

Line 245: remove comma

Line 285: please cite some primary reference papers in the HTS assay section aside from the single review

Line 302: cite more of the primary literature in this section

Line 338: should be "culture" instead of "cultures"

Line 339: rephrase to "and the knowns and unknowns"

Line 347: clarify this sentence: it sounds like the authors are indicating NS5 is not present during mouse infection!

Line 379: comma at the end of the line needed

Line 400: "on" should be "one"

Line 422: move "especially" to after "are"

Line 454: should be "behind-the-scenes"

Line 525: commas are needed after "MCMV", "HCMV" and "demonstrated"

Line 533: comma after "(pNPC)"

Lines 577-603: Studies conducted in the last few years have looked at host markers, such as the neural stem cell marker SOX2, as factors defining ZIKV tropism (in the case of SOX2, it has been shown to be negatively correlated with the interferon response). Please discuss host-side genetic factors that could be involved in the three viruses you discuss in this review, as this is an important "other side of the coin" that compliments viral genetic diversity.

Line 637: rephrase: "Pluripotent stem cell-based in vitro models grow increasingly complex."

Line 659: comma after "turn"

Line 661: comma after "study"

Line 662: change "cells. They" to "cells, but they"

Line 687: change "host. They" to "host, but they"

Line 701: "extent" should be "extend"

Line 703: "to a big extent" should be "largely"

Line 704: "on" should be "of", second "on" should be "from"

Author Response

General comments and general changes made to the original manuscript

The authors want to thank the reviewers for their time spent on our manuscript. Their helpful comments and suggestions for revision improved the manuscript.

Point-by-point response to the reviewer’s suggestions

Reviewer 2

Furthermore, I have a major concern with the inclusion of unpublished, non-peer reviewed data (line 544-546, Figure 3) from the authors themselves in the manuscript. This should not be present in a balanced review article of the published literature in the field, and should be removed in the final version of the paper.

# The primary, non-peer reviewed data was removed as suggested. The entire Figure 3 was omitted to avoid an over emphasis on data on rubella virus and to shorten the manuscript as suggested by reviewer 1.

General comment: The authors frequently refer the reader to external reviews, usually in a passing sentence, which is not great form for a review paper. While this is OK to do a few times, it is done too frequently in this paper (there are more instances than the two listed in the specific comments). Either delete some of these sections, or cite the primary literature and expand on these ideas.

# In our efforts to shorten the manuscript we have deleted several sentences with a reverence to external reviews.

Specific comments:

Line 32: should be ": it [...]"

# Changed as suggested.

Line 34: no comma

# As suggested, the comma was deleted.

Line 67-68: reorganize the text to have a discussion of the role of interferons in viral pathogenesis before the discussion of IFNAR knockout mice (to provide a logical flow for the non-expert)

# The paragraph was restructured as suggested.

Line 84: "advance. Teratogenic [...]" should be "advance, but teratogenic [...]"

# Changed as suggested.

Line 87: rephrase sentence so that it is not a fragment

# As also suggested by reviewer 1, the sentence was revised as follows: Transplacental transmission, the crosstalk at the embryo/fetal-maternal compartment and the mode of impairment of early embryonal development remain not only for RV ill-defined.

Line 92: "un-accessible" should be "inaccessible"

# Changed as suggested.

Line 113: rephrase: "The placenta acts as an efficient barrier against pathogens."

# Changed as suggested.

Line 115: comma after "family", "which were" should be "which are"

# Changed as suggested.

Line 128: "on" should be "onward"

# Changed as suggested.

Line 176: comma after "viremia"

# The comma was added as suggested.

Line 233: define "CRS" acronym in the text here

# This point was also noted by reviewer 1. The abbreviation was already introduced in line 211. We have revised the sentence in 211, such that the term CRS is now introduced in line 268 in the context of the disease.

Line 245: remove comma

# The comma was removed as suggested.

Line 285: please cite some primary reference papers in the HTS assay section aside from the single review

# Primary literature including a description of the pan-caspase inhibitor Emricasan was added (line 335 to 344),

Line 302: cite more of the primary literature in this section

# As primary literature we have added two additional publications on Zika virus strains and removed some of the publications on global surveillance of rubella. However, the cited manuscripts on rubella virus epidemiology are not review papers and reflect the lack of available primary data. Thus, publications are based on interpretation of data available in data banks.

Line 338: should be "culture" instead of "cultures"

# Changed as suggested.

Line 339: rephrase to "and the knowns and unknowns"

# Changed as suggested.

Line 347: clarify this sentence: it sounds like the authors are indicating NS5 is not present during mouse infection!

# The authors are grateful for this comment, as the sentence was indeed misleading. We have revised the sentence accordingly.

Line 379: comma at the end of the line needed

# Changed as suggested.

Line 400: "on" should be "one"

# Changed as suggested.

Line 422: move "especially" to after "are"

# Changed as suggested.

Line 454: should be "behind-the-scenes"

# Reviewer 1 also commented on this phrase, which we have deleted.

Line 525: commas are needed after "MCMV", "HCMV" and "demonstrated"

# Changed as suggested.

Line 533: comma after "(pNPC)"

# Changed as suggested.

Lines 577-603: Studies conducted in the last few years have looked at host markers, such as the neural stem cell marker SOX2, as factors defining ZIKV tropism (in the case of SOX2, it has been shown to be negatively correlated with the interferon response).

# The authors are grateful for pointing out SOX2 as an example for another host factor that defines ZIKV tropism. We have added this aspect to the manuscript and cited the publication by Zhu et al. in Cell Stem Cell (doi 10.1016/j.stem.2019.11.016) accordingly.

Please discuss host-side genetic factors that could be involved in the three viruses you discuss in this review, as this is an important "other side of the coin" that compliments viral genetic diversity.

# This important aspect was also highlighted by reviewer 1. We have added primary literature on examples of the contribution of host genetics of congenital virus infections (line 533 to line 566).

Line 637: rephrase: "Pluripotent stem cell-based in vitro models grow increasingly complex."

# Changed as suggested.

Line 659: comma after "turn"

# Changed as suggested.

Line 661: comma after "study"

# Changed as suggested.

Line 662: change "cells. They" to "cells, but they"

# Changed as suggested.

Line 687: change "host. They" to "host, but they"

# Changed as suggested.

Line 701: "extent" should be "extend"

# Changed as suggested.

Line 703: "to a big extent" should be "largely"

# Changed as suggested by reviewer 1 to large extent.

Line 704: "on" should be "of", second "on" should be "from"

# Changed as suggested.

Round 2

Reviewer 1 Report

The authors have addressed the reviewers comments. A final suggestion would be to change the title to "Pluripotent stem cell-based models: a window into virus infections during early pregnancy".

Reviewer 2 Report

The authors have addressed each of my comments thoroughly, and I recommend publication of the paper.